# Carbon Storage and Sequestration Analysis by Urban Park Grid Using i-Tree Eco and Drone-Based Modeling

**Juhyeon Kim [1], Youngeun Kang [1,*], Dongwoo Kim [2], Seungwoo Son [3] and Eujin Julia Kim [4,*]** 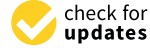

[1] Department of Landscape Architecture, Gyeongsang National University, Jinju 52725, Republic of Korea; qlwm19@naver.com
[2] Gyeonggi Research Institute, Suwon 16207, Republic of Korea; dwkim@gri.re.kr
[3] Korea Environment Institute, Sejong 30147, Republic of Korea; swson@kei.re.kr
[4] Department of Environmental Landscape Architecture, Gangneung-Wonju National University, Gangneung 25457, Republic of Korea
[*] Correspondence: yekang@gnu.ac.kr (Y.K.); ejkim@gwnu.ac.kr (E.J.K.); Tel.: +82-55-772-3301 (Y.K.); +82-33-640-2478 (E.J.K.)

**Abstract:** Urban areas play a crucial role in carbon absorption, while also producing a considerable amount of carbon emissions. However, there has been a lack of research that has systematically examined the carbon storage and sequestration in green spaces located within urban environments, at a spatial scale. This study analyzes carbon storage and sequestration in Yurim Park, Daejeon, South Korea on a grid basis to fill the research gap. The research compares the variation in sequestration capacity across different grids and provides insights into the development of sustainable urban parks in urban planning. The classification of grids is based on specific site characteristics, such as land cover, tree distribution, type, and density. This results in a total of seven distinct types. The study employs a combination of the I-tree eco model, drone-based modeling, and on-site surveys to estimate carbon storage and sequestration in urban parks. The results show that the average carbon storage per unit area in the entire park was 15.3 tons of carbon per hectare, ranging from a minimum of 5.0 to a maximum of 21.4 tons per hectare. For the planted area, the average carbon storage was 8.6 tons per hectare. Grids with green areas dominated by broad-leaved trees and closed canopy cover had the highest carbon sequestration and storage values. The planting area ratio and the type of trees planted were found to directly influence the carbon sequestration capacity per unit area of urban parks. This study stands out from previous research by conducting a detailed area-based comparison and analysis of carbon sequestration capacity in urban parks using sophisticated measurement techniques. The findings offer direct insights into strategies and policies for securing future urban carbon sinks and can be of practical use in this regard.

**Keywords:** carbon neutral; carbon storage; carbon sequestration; i-Tree Eco; UAV; urban green space planning



## 1. Introduction

The world is currently confronting the challenge of climate change, and it is anticipated that its impacts on human societies will intensify in the years to come [1–4]. Evidence has shown that human activities are a direct source of carbon dioxide, a significant greenhouse gas that contributes to climate change [3,5]. The 48th General Assembly of the Intergovernmental Panel on Climate Change (IPCC) stated that in order to limit the global average temperature rise to 1.5 degrees Celsius, global carbon dioxide emissions must be reduced by at least 45 percent compared to 2010 levels [6]. Furthermore, the report emphasized the need to achieve net zero carbon neutrality by 2050. Carbon neutrality involves two main strategies: carbon reduction, which aims to lower carbon emissions, and carbon sink, which focuses on capturing or absorbing emitted carbon [7]. While initial efforts in carbon neutrality policies were predominantly focused on reducing emissions, there is now a

growing shift towards actively enhancing carbon sinks. This broader approach includes measures such as managing forests and reforestation, implementing agricultural management practices, conserving wetlands and marine ecosystems, as well as the deployment of carbon capture technologies [8–11].

Urban Green Spaces (UGS) play a crucial role in absorbing carbon, which helps to reduce greenhouse gas levels [12–15]. These areas are essential for mitigating climate change and offer significant environmental benefits, such as reducing pollution and improving water quality [16–19]. The importance of carbon sinks is increasingly being recognized, and efforts are being made to not only expand them but also optimize their carbon absorption efficiency. Several studies [20,21] indicate that urban regions, which generally have high carbon dioxide emissions, can be effectively utilized to enhance carbon absorption capabilities. This suggests that a strategic approach to urban planning and environmental management could be employed to address this issue. According to a recent study [22], there is a 95% probability that 68% of the world's population lives in urban areas. It is also predicted that urban activities will be responsible for 71% of carbon dioxide emissions linked to energy consumption [23,24]. Therefore, it is crucial to improve the ability of UGS to absorb carbon, which is essential for effectively balancing carbon emissions [25]. South Korea has established a plan to achieve carbon neutrality by 2050, which includes increasing urban carbon absorption by expanding UGS. The national goal for reducing greenhouse gas emissions by 2030 aims to have forests and UGS account for 96% of the overall carbon sink objectives. Therefore, devising and implementing related strategies becomes indispensable.

Research on carbon sinks has used various methods to measure carbon storage and sequestration abilities. These methods include field surveys, National Forest Inventories (NFI), remote sensing technology, geostatistical techniques, and modeling based on environmental factors [26–28]. While field surveys are thorough, they can be expensive and time-consuming [29]. Remote sensing techniques face challenges such as inconsistencies in spatial sampling location, shadowing effects, and variability in spectral responses [30,31]. The development of remote sensing technologies, such as LiDAR, drone-based modeling, and machine learning, has improved the accuracy of measuring carbon sinks in urban environments [29,32,33]. Compared to forested areas, urban environments are more complex in structure, making it challenging to measure carbon sinks accurately. However, current research mainly concentrates on forests, and there is limited exploration of the potential for carbon storage and sequestration within urban carbon sinks. Moreover, research into the carbon storage and sequestration capabilities of urban greenery and parks has predominantly targeted the carbon sequestration potential of individual trees rather than assessing collective green areas [34,35]. This narrow focus has limited the ability to develop comprehensive spatial planning or inform policy in this area.

To fill this research gap, this study aims to measure the capacity of various types of plants to absorb carbon in urban parks in Daejeon, South Korea. The study will provide spatial planning recommendations based on the findings. The research will be conducted as follows: First, carbon storage and sequestration will be calculated and compared by area within the park and analysis space type. Second, factors affecting carbon absorption and sequestration by zone and individual trees within the park will be analyzed. Third, planting guidelines will be created based on the results to increase the efficiency of carbon sinks in urban parks. This research will improve the methodology for measuring carbon storage and sequestration in urban areas, enhance the effectiveness of carbon neutral policies, and contribute to creating sustainable cities.

## 2. Literature Review

### 2.1. The Concept of Carbon Storage and Carbon Sequestration

The storage of carbon in trees plays a vital role in maintaining a sustainable environment and offsetting carbon emissions. Trees are highly efficient at storing carbon by absorbing carbon dioxide ($CO_2$) from the atmosphere through photosynthesis. This process

converts $CO_2$ into glucose and oxygen using solar energy. The storage of carbon in trees is an active process that allows them to buffer climate fluctuations and support long-term growth. The quantity of carbon stored in trees is influenced by various factors such as biomass structure, species composition, canopy cover, and tree density [36].

Carbon storage includes a variety of processes and methods that hold or contain carbon in different reservoirs such as the atmosphere, oceans, land, and biomass. In forests, carbon storage is usually classified into four types: tree biomass, soil organic carbon (SOC), dead and dying trees, and dead wood [37]. It is important to note that this classification is not exhaustive, and other types of carbon storage may also exist. Carbon storage refers to both carbon retention and carbon sequestration through absorption. However, studies on carbon storage in trees have often separated carbon storage and carbon sequestration into distinct metrics [38–40]. The carbon storage concept pertains to the quantity of carbon contained within tree biomass.

Carbon sequestration is the process of removing carbon from the atmosphere and storing it in carbon sinks such as biomass. It involves both the current carbon storage by biomass and the process of carbon sequestration through the seasonal growth of trees. Recent studies have addressed both aspects of carbon storage and carbon sequestration.

Carbon storage is the total amount of carbon that a tree has accumulated over many years of growth. This is the amount of carbon stored in the form of carbon dioxide through photosynthesis, which is the process of synthesizing organic matter necessary for tree growth. The carbon storage is calculated by deriving the dry weight of the tree. The carbon storages of woody biomass are calculated by multiplying it with the IPCC's carbon conversion factor (0.5). This value is then multiplied by the carbon dioxide conversion factor (44/12) to obtain the carbon dioxide sequestration of the tree at the time of measurement, as per the National Academy of Forestry Sciences (2012).

The annual carbon sequestration or sequestration of carbon by a tree during a year can be calculated by subtracting the estimated carbon storages in year X + 1 from those in year X [41]. Gross carbon sequestration includes annual mortality and decay, while net carbon sequestration excludes these factors [42].

### 2.2. Research on Measuring Carbon Sequestration and Carbon Storage in Urban Parks

Within the fields of urban planning and environmental sustainability, the significant role of urban parks in the processes of carbon storage and sequestration has become a key focus of research [8,14]. This growing interest aligns with global efforts to combat climate change by positioning green urban areas as a crucial element in reducing atmospheric carbon dioxide levels. The comprehensive literature on this topic covers both the biological and physical aspects of carbon sequestration and extends to examine the complex interrelations between the design of urban green spaces and their wider ecological consequences.

Tian and An [43] have provided foundational insights, concerning the carbon storage abilities of park vegetation through the application of biomass evaluation methods to determine the structure and variety of trees and shrubs. This focus on species-specific carbon sequestration highlights the literature's emphasis on quantitative assessments of carbon dynamics, taking into account elements such as growth rates and the sequestration potential of different tree species.

Building on these insights, more recent studies, Shen et al. [44] expanded the research framework to measure urban forest biomass by comparing historical and contemporary data, leveraging information from multiple forest plots and Landsat imagery. Their findings demonstrate a decrease in carbon density from urban centers to suburban regions, emphasizing the importance of such data for the efficient management and optimization of carbon sequestration in urban forests.

Fan et al. [45] have examined the carbon sequestration potential of urban forests, underlining the importance of factors such as canopy covers and tree height. Currently, Jin et al.'s [46] investigations into the effect of tree species selection on carbon sequestration

potential have revealed the significance of Photosynthetic rate (*Pn*), crown area (*CA*), and leaf area index (*LAI*) in the carbon sequestration process.

Research by Shadman et al. [21] on how urban park design influences microclimate and carbon sequestration, alongside [24]'s comprehensive analysis of the spatial characteristics of green spaces and their carbon sequestration capacities, highlights the complex link between biodiversity, forest structures, and carbon density, emphasizing the importance of scale in these relationships. Furthermore, to understand the impact of land use changes on carbon sequestration efficiency, a study in Beijing explored the factors affecting carbon sequestration in urban green spaces and proposed strategies for sustainable urban green space planning [47]. The use of GIS (Geographic Information System) and remote sensing by [47] to study green space configuration represents the application of sophisticated methods to optimize urban green areas for carbon sequestration.

A global meta-analysis by [48], which quantifies the carbon sequestration contributions of urban trees, encapsulates the environmental impact of urban forestry practices, offering a global perspective on the variables affecting urban tree carbon dynamics, from environmental conditions to management strategies.

This study aims to fill the noted gaps in existing literature, especially regarding the direct link between planting design and its effects on carbon storage and sequestration. By weaving together empirical findings from prior research with an in-depth analysis of planting design's impact on carbon dynamics, this study seeks to reveal urban parks' hidden capabilities as sustainable instruments posed by the climate change crisis. This effort is crucial not just for enhancing urban resilience to climate change, but also for laying the groundwork for evidence-based practices in urban planning and landscape architecture, aimed at optimizing the ecological benefits of urban green spaces. Through this approach, the study contributes to a refined understanding of urban parks' multifaceted roles within urban ecosystems, steering the creation of sustainable, resilient, and carbon-efficient urban landscapes.

## 3. Materials and Methods

### 3.1. Conceptual Framework

In this study, based on a review of carbon storage and sequestration by trees, the carbon storage and sequestration of urban parks was calculated by tree species, study area, and analysis area type using the i-Tree Eco model (ver.6.0, United States Department of Agriculture, USDA, Washington, DC, USA). The i-Tree Eco model used in this study requires an accurate understanding of the current status of park trees. After conducting a field survey, the results of carbon storage and sequestration were obtained using the i-Tree Eco model, and planting strategies that can increase carbon sequestration in urban parks are proposed (Figure 1).

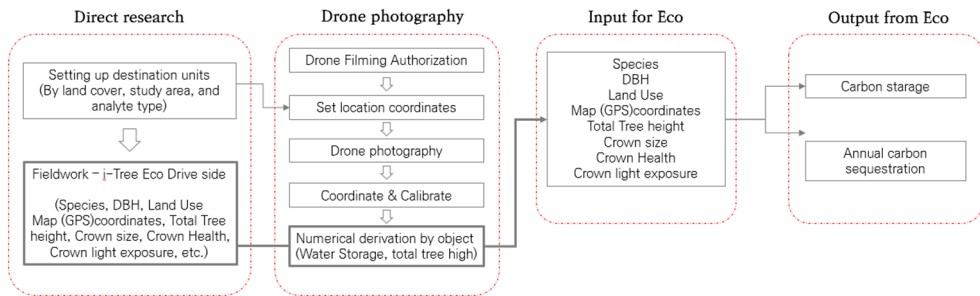

**Figure 1.** Conceptual framework.

### 3.1.1. Study Area

Daejeon Metropolitan City is subdivided into five administrative districts: Yuseong-gu, Seo-gu, Daedeok-gu, Jung-gu, and Dong-gu. Among these districts, Yuseong-gu boasts the highest number of parks, constituting 36% of the total within Daejeon Metropolitan City [49]. In the context of this study, the examination focused on urban parks situated

within Yuseong-gu, Daejeon, South Korea, specifically homing in on those characterized by waterside spaces. The selection criteria deliberately excluded parks with existing forests, aiming to streamline the classification of park types. The methodology employed for conducting the vegetation survey of these urban parks entailed a thorough assessment of whether park data was made publicly available, prioritizing considerations for on-site accessibility. This approach was crucial for obtaining a precise understanding of the current status of each park under investigation. Moreover, to ensure accurate measurements of both tree height and crown height, the study site was selected with careful consideration, giving weight to the integration of drone photography.

This study focuses on Yulim Park, which is located in Yuseong-gu, Daejeon, South Korea, as shown in Figure 2. The park covers a vast area of 57,400 square meters, out of which 36,583 square meters are dedicated to green spaces, which makes up approximately 63.7% of the park's total area. The built-up area of the park is 20,817 square meters, which accounts for around 26.3% of the park's total area, and it includes a unique peninsula that extends over 2600 square meters. Yulim Park boasts a diverse collection of flora, including 64,082 seasonal trees and 135,450 perennial plants.

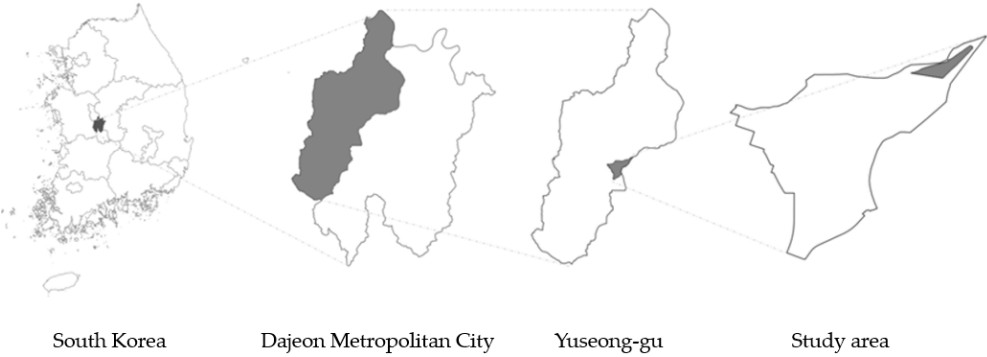

South Korea      Dajeon Metropolitan City      Yuseong-gu      Study area

**Figure 2.** Study area.

### 3.1.2. Spatial Classification Methodology

The study area was divided into 20 m × 20 m grids using QGIS (3.22.15) to categorize the analytical spaces (Figure 3). The biotope classification framework established in earlier studies [38,50] was used to classify the analysis space types, taking into account the distinct carbon sequestration efficiencies of different biotopes. The framework was adjusted as necessary to enhance its applicability to our study. The previous study identified nine types based on land cover, vegetation cover, and dominant vegetation. Land cover classifications included grey, green, and blue spaces, determined by the highest percentage in each area. Vegetation cover was categorized into open, partially open, and closed spaces. Dominant vegetation types comprised grass, shrub, evergreen, and deciduous trees.

This study followed the classification type of previous studies but modified the detailed criteria to apply to this study. The criteria were adjusted to suit the specific requirements of this research.

Firstly, grassland areas were classified into two categories: grassland and waterfront areas, based on the land cover map. Secondly, areas were classified as open, partially open, or closed based on vegetation cover, as determined by satellite imagery. If the vegetation cover is less than 30%, the area is classified as open. If the vegetation cover is between 30% and 70%, the area is classified as partially open. If the vegetation cover is more than 70%, the area is classified as closed. The classification of grassland areas is based on the percentage of vegetation cover. The waterfront area is mainly planted with herbaceous species and is classified as open if the vegetation cover is less than 20% and partially open if it is more than 20%. These criteria differ from those used for grassland areas. The dominant plant species in Yurim Park were categorized based on the proportion of deciduous and evergreen trees in the area using the as-built drawings (Table 1).

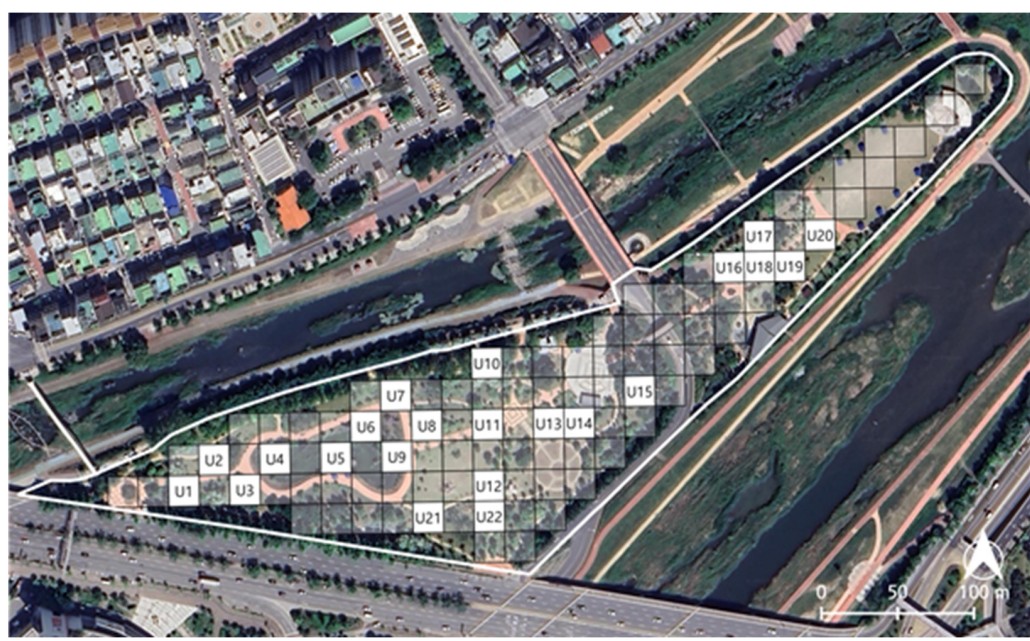

**Figure 3.** Study area grid.

**Table 1.** Flowchart for analytics space typing.

| Land Cover | Tree Coverage | Dominant Plant Category | Type | Unit |
|---|---|---|---|---|
| Green space | Open | Mainly evergreen trees | G-E-E | U1, U10, U15 |
| | Open | Mainly deciduous trees | G-E-D | U16 |
| | Partly open | Mainly evergreen trees | G-P-E | U3, U14, U17 |
| | Partly open | Mainly deciduous trees | G-P-D | U8, U19, U20, U22 |
| | Closed | Mainly evergreen trees | G-O-E | U7, U12, U13 |
| | Closed | Mainly deciduous trees | G-O-D | U2, U11, U18, U21 |
| Blue space | | | B-P | U6, U9 |
| | | | B-O | U4, U5 |

*3.2. Field Survey and Drone Modelling*

The survey area for this study was identified using satellite images of the park, and field surveys were conducted on 22 June 2023, and 6 July 2023 to determine the status of trees in the study area.

The i-Tree Eco model used in this study relies on survey data collected through thorough field surveys. The i-Tree Eco model requires essential data such as the coordinates of tree locations, tree species, diameter at breast height (cm), basal height (m), crown width (m), crown loss rate (%), tree health (%), and crown light exposure (1 to 5). This foundational information drives the comprehensive analysis of the ecological and physiological aspects of the trees within the study area.

Drone filming was conducted to capture ortho imagery and the vegetation Digital Surface Model (DSM) of the study area. A flight path with a 90% overlap was established at an altitude of 80 m, resulting in the acquisition of 929 images covering the entire study area. Ground surveys were conducted on 17 ground control points installed prior to filming to enhance the location accuracy of the captured images. The study area's vegetation distribution, ground cover (in meters), and crown height (in meters) were determined through location correction using Pix4dmapper (Prilly, Switzerland) to construct orthoimages and the DSM. The study area's vegetation distribution, ground cover (in meters), and crown

height (in meters) were determined through location correction using Pix4dmapper to construct orthoimages and the DSM.

Using QGIS 3.16, the data collected from field surveys on tree information, as well as the tree height and crown height data obtained through drone imaging, were mapped and visually presented for further analysis and interpretation.

### 3.3. i-Tree Model

In this study, i-Tree Eco modeling was used to estimate the carbon storage and sequestration capacity of urban park vegetation. The i-Tree Eco program (version 6.0.32) was used to quantify and evaluate the environmental value of urban forests and trees for carbon storage and sequestration. The program uses local meteorological information and field and inventory data to provide information on the environmental performance of tree stands, including carbon storage and sequestration. Notably, South Korea has been added as a supported region starting in 2019. The data required to run the i-Tree Eco model is tree diameter and species, while the input of land use and tree canopy information provides accurate information on carbon storage and sequestration. Tree canopy information includes canopy loss rate, health status, canopy width, and canopy light exposure in five directions.

The tree species studied were *Acer buergerianum*, *Acer palmatum*, *Aesculus turbinata*, *Chionanthus retusa*, *Cornus kousa*, *Cornus officinalis*, *Diospyros kaki*, *Ginkgo biloba*, *Lagerstroemia indica*, *Pinus densiflora*, *Pinus densiflora f. multicaulis*, *Pinus koraiensis*, *Prunus sargentii*, *Prunus yedoensis*, *Quercus palustris*, *Quercus serrata*, *Salix pseudolasiogyne*, and *Taxodium distichum*. By matching the i-Tree Eco library to the studied tree species, in case of mismatch at the species level, data were obtained at the higher genus level and analyzed in the following classifications (Table 2).

**Table 2.** Tree species list matching i-Tree Eco library.

| No. | Name of Species | i-Tree Eco Library |
|:---:|:---:|:---:|
| 1 | *Acer buergerianum* | *Acer buergerianum* |
| 2 | *Acer palmatum* | *Acer palmatum* |
| 3 | *Aesculus turbinata* | *Aesculus turbinata* |
| 4 | *Chionanthus retusa* | *Chionanthus retusus* |
| 5 | *Cornus kousa* | *Cornus kousa* |
| 6 | *Cornus officinalis* | *Cornus officinalis* |
| 7 | *Diospyros kaki* | *Diospyros kaki* |
| 8 | *Ginkgo biloba* | *Ginkgo biloba* |
| 9 | *Lagerstroemia indica* | *Lagerstroemia indica* |
| 10 | *Pinus densiflora, Pinus densiflora f. multicaulis* | *Pinus densiflora* |
| 11 | *Pinus koraiensis* | *Pinus koraiensis* |
| 12 | *Prunus sargentii* | *Prunus sargentii* |
| 13 | *Prunus yedoensis* | *Prunus x yedoensis* |
| 14 | *Quercus palustris* | *Quercus palustris* |
| 15 | *Quercus serrata* | *Quercus serrata* |
| 16 | *Salix pseudolasiogyne* | *Salix pseudolasiogyne* |
| 17 | *Taxodium distichum* | *Taxodium distichum* |

### 3.4. Statistical Methodology

In this study, the survey area was identified through satellite images of the park during the field survey. Field surveys were conducted on 22 June 2023, and 6 July 2023 to identify the status of trees in the study area in the park. analysis utilized Microsoft

Office Excel 2016 and statistical software SPSS 29.0.1.0 (IBM Corp. IBM SPSS Statistics for Windows, Armonk, NY, USA) for Windows to examine the impact of various tree characteristics (crown diameter, crown width, biomass, crown health, and crown loss rate) on carbon sequestration as determined by the i-Tree Eco model. Tree species with limited representation were excluded from the study. Factor analysis was conducted on the carbon sequestration of eight specific tree species: *Acer palmatum*, *Chionanthus retusa*, *Cornus officinalis*, *Lagerstroemia indica*, *Pinus densiflora*, *Pinus densiflora f. multicaulis*, *Prunus yedoensis*, and *Quercus palustris* (Table 3). In this analysis, the carbon sequestration of each tree species served as the dependent variable, while tree information such as crown diameter, crown width, biomass, crown health, and crown loss rate acted as independent variables. To calculate the biomass value, the carbon storage value of each individual tree was multiplied by 0.4. The dependent variable was normalized carbon sequestration for each tree. Both linear and non-linear regression approaches were iteratively employed to determine the most suitable regression equation and variables. Given the proportional relationship between carbon storage and biomass, the analysis focused solely on regressions related to carbon sequestration.

**Table 3.** Species used in statistical analysis.

| Tree Species | Number of Objects |
|---|---|
| *Acer palmatum* | 14 |
| *Chionanthus retusus* | 72 |
| *Cornus officinalis* | 10 |
| *Lagerstroemia indica* | 7 |
| *Pinus densiflora* | 40 |
| *Pinus densiflora f. multicaulis* | 11 |
| *Prunus yedoensis* | 7 |
| *Quercus palustris* | 18 |

## 4. Results

### 4.1. Carbon Storage and Carbon Sequestration (CSCS) of Study Area

Total carbon storage by species was found to be highest for *Pinus densiflora* and *Pinus densiflora f. multicaulis*, followed by *Chionanthus retusa*, *Quercus palustris*, *Prunus yedoensis* and *Lagerstroemia indica* (Table 4). When looking at carbon storage per individual tree, *Taxodium distichum* had the highest carbon storage at 2679.1 kgC/tree, followed by *Salix pseudolasiogyne* at 2770.7 kgC/tree, and *Ginkgo biloba* at 1900.6 kgC/tree. The analysis showed that deciduous trees account for a significant proportion of carbon storage.

**Table 4.** Carbon storage and sequestration by tree species.

| i-Tree Eco Tree Species | Number of Objects | Carbon Storage (kg) | Carbon Sequestration (kg/year) |
|---|---|---|---|
| *Acer buergerianum* | 1 | 594.9 | 3.1 |
| *Acer palmatum* | 14 | 412.4071 | 46.2 |
| *Aesculus turbinata* | 1 | 978.1 | 0.6 |
| *Chionanthus retusus* | 72 | 469.5708 | 818.8 |
| *Cornus kousa* | 2 | 192.85 | 19.8 |
| *Cornus officinalis* | 10 | 63.1 | 77.2 |
| *Diospyros kaki* | 1 | 91.5 | 8.2 |
| *Ginkgo biloba* | 1 | 1900.6 | 24.4 |
| *Lagerstroemia indica* | 7 | 459.8286 | 45.1 |
| *Pinus densiflora* | 51 | 926.4922 | 1085.7 |
| *Pinus koraiensis* | 1 | 914.1 | 0.5 |
| *Prunus sargentii* | 1 | 294.2 | 24.6 |
| *Prunus x yedoensis* | 7 | 815.9 | 85.2 |
| *Quercus palustris* | 18 | 588.6556 | 533.7 |
| *Quercus serrata* | 1 | 1250.7 | 34.6 |

**Table 4.** *Cont.*

| i-Tree Eco Tree Species | Number of Objects | Carbon Storage (kg) | Carbon Sequestration (kg/year) |
|---|---|---|---|
| *Salix pseudolasiogyne* | 1 | 1970.7 | 65.6 |
| *Taxodium distichum* | 1 | 2679.1 | 25.5 |
| Total | 190 | 118,050.4 | 2898.8 |

The carbon sequestration of each tree species In the study area was analyzed using the i-Tree Eco model, and the highest carbon sequestration was found for *Pinus densiflora* and *Pinus densiflora f. multicaulis*, followed by *Chionanthus retusa*, *Quercus palustris*, and *Prunus yedoensis* (Table 4). When analyzing the carbon sequestration per individual tree, *Salix pseudolasiogyn* had the highest carbon sequestration at 65.6 kgC/year/tree, followed by 34.6 kgC/tree, and the third highest at 29.7 kgC/tree.

*4.2. The Difference of CSCS by Grid*

In this comprehensive study, which included 22 designated study areas labeled U1 through U22, the i-Tree Eco model meticulously calculated carbon storage for each specific study area. The analysis identified U15, U22, U8, U1, and U10 as having particularly high levels of carbon storage (Table 5). These areas are characterized by the prevalence of high-density trees, coupled with significant stem diameter, stem width, and height parameters, which are closely related to tree biomass.

**Table 5.** Carbon storage and sequestration by grid.

| Grid | Carbon Storage (kg) | Carbon Sequestration (kg/year) | Grid | Carbon Storage (kg) | Carbon Sequestration (kg/year) |
|---|---|---|---|---|---|
| U1 | 9636.50 | 245.10 | U12 | 2446.50 | 128.50 |
| U2 | 5807.70 | 203.80 | U13 | 5256.50 | 150.70 |
| U3 | 4912.70 | 44.10 | U14 | 7967.10 | 106.90 |
| U4 | 1970.70 | 65.6 | U15 | 13,422.00 | 193.00 |
| U5 | 294.2 | 24.6 | U16 | 7609.60 | 258.10 |
| U6 | 3271.60 | 100.40 | U17 | 3817.30 | 56.80 |
| U7 | 2138.40 | 108.00 | U18 | 6693.30 | 108.20 |
| U8 | 10,025.50 | 84.70 | U19 | 2957.20 | 106.80 |
| U9 | 1453.30 | 59.70 | U20 | 4687.40 | 225.90 |
| U10 | 9175.50 | 131.60 | U21 | 1311.10 | 95.40 |
| U11 | 1559.60 | 92.20 | U22 | 11,636.70 | 292.80 |
| - | - | - | Total | 118,050.40 | 2898.80 |

Shifting the focus to carbon sequestration, the study highlighted U22, U16, U1, U20, and U2 as having the highest levels of this critical ecological process. In particular, U22 and U1 emerged as areas with both elevated carbon storage and sequestration (Table 5). A common feature characterizing these areas is the substantial presence of planted trees, highlighting the influential role of tree density in the carbon dynamics of urban environments.

Figures 4 and 5 represent carbon storage and sequestration by grid location, respectively. The darker the color, the darker the storage and absorption amount, and the natural break function of GIS was used to distinguish the interval. These figures indicate that units with substantial carbon storage also exhibit high levels of carbon sequestration, though variations in storage and sequestration rates across different grids can be attributed to the presence of certain tree species known for carbon sequestration capabilities.

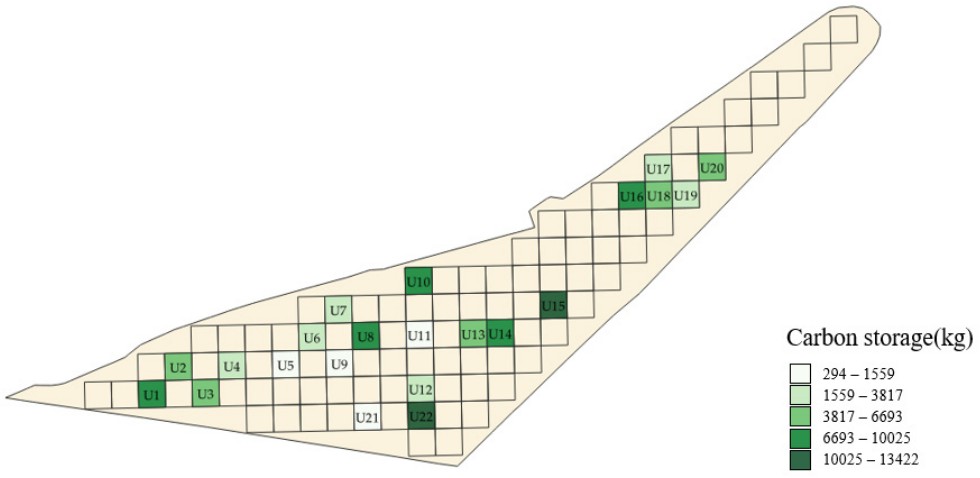

**Figure 4.** Carbon storage by grid location (GIS).

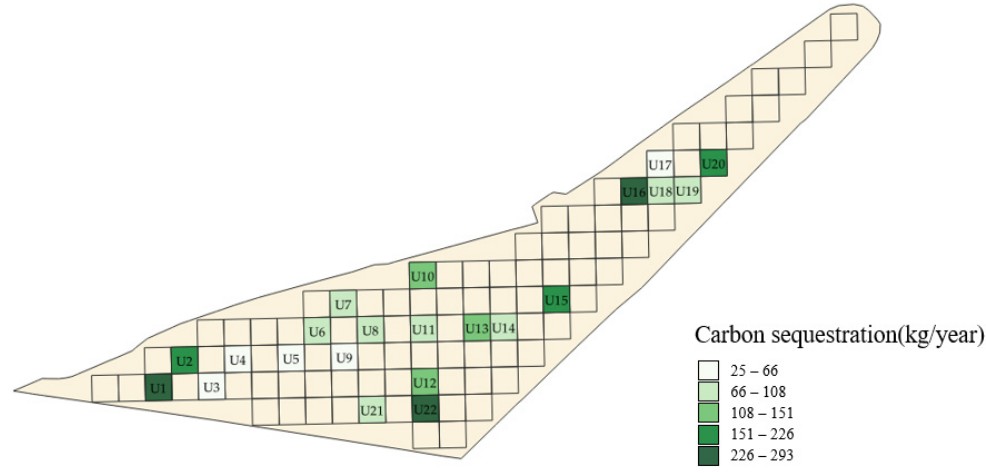

**Figure 5.** Carbon sequestration by grid location (GIS).

*4.3. The Difference of CSCS by Analysis Area Type*

The carbon storage and sequestration by type of analyzed space was obtained as the total carbon storage and sequestration by type per unit area. The space with the highest carbon storage is the G-E-E type with 10,744.67 kg/m². The second is the G-E-D type with 7609.60 kg/m², and the third is the G-P-D type with 7326.7 kg/m² of carbon (Table 6).

**Table 6.** Carbon storage and sequestration by study area type.

| Type | Area | Carbon Storage (kg/m²) | Carbon Sequestration (kg/m²/year) |
|---|---|---|---|
| G-E-E | U1, U10, U15 | 10,744.67 | 189.9 |
| G-E-D | U16 | 7609.60 | 258.1 |
| G-P-E | U3, U14, U17 | 5565.7 | 69.26667 |
| G-P-D | U8, U19, U20, U22 | 7326.7 | 177.55 |
| G-O-E | U7, U12, U13 | 3280.467 | 129.0667 |
| G-O-D | U2, U11, U18, U21 | 3842.925 | 124.9 |
| B-P | U6, U9 | 2362.45 | 80.05 |
| B-O | U4, U5 | 1132.45 | 45.1 |

The lowest carbon sequestration compared to carbon storage is in the G-E-E and G-P-E types, which have relatively more evergreen trees than other types (Table 6). When looking at carbon storage and sequestration in the study area, the common characteristics of the types that favor carbon sequestration efficiency are G-E-D and G-P-D, which are grasslands with deciduous trees as the dominant species.

*4.4. Tree Attributes Affecting CS*

4.4.1. Tree Characteristics That Affect Carbon Sequestration in Maple Trees

The model itself was analyzed as significant with a significance probability of 0.000, and the explanatory power of the factor model affecting maple carbon sequestration was 78.1%.

Finally, the coefficient analysis showed that the only significant variable affecting the amount of carbon sequestration in maple trees was the width of the tree crown, which had a negative relationship with carbon sequestration. The regression equation can be expressed as y = −0.007(crown width) + 10.413 (Table 7).

**Table 7.** Analysis of factors affecting carbon sequestration in maple trees.

| Model | Coefficients [a] | | | *t* | Significance |
|---|---|---|---|---|---|
| | Unstandardized Coefficients | | Standardized Coefficients | | |
| | B | Std. Error | Beta | | |
| 1 (Constant) | 10.413 | 1.166 | | 8.927 | 0.000 |
| Crown Width | −0.007 | 0.001 | −0.884 | −6.544 | 0.000 |

[a] Dependent Variable: Carbon sequestration. R Square = 0.781.

4.4.2. Tree Characteristics That Affect Carbon Sequestration in *Quercus palustris*

The significance probability of the model itself was found to be significant at 0.000, and the explanatory power of the factor model influencing the carbon sequestration of *Q. palustris* Muench was found to be high at 95.7%.

In the coefficient analysis, it was found that biomass and crown missing significantly affect carbon sequestration in red oak. Biomass was positively related to carbon sequestration and negatively related to crown missing. The regression equation is y = 0.012(Biomass) − 0.127(crown missing) + 14.483 (Table 8).

**Table 8.** Analysis of factors affecting carbon sequestration in *Quercus palustris*.

| Model | Coefficients [a] | | | *t* | Sig. |
|---|---|---|---|---|---|
| | Unstandardized Coefficients | | Standardized Coefficients | | |
| | B | Std. Error | Beta | | |
| 1 (Constant) | 14.483 | 2.241 | | 6.463 | 0.000 |
| Biomass | 0.012 | 0.001 | 0.789 | 10.473 | 0.000 |
| Crown Missing | −0.127 | 0.039 | −0.248 | −3.288 | 0.005 |

[a] Dependent Variable: Carbon sequestration. R Square = 0.957.

4.4.3. Tree Characteristics That Affect Carbon Sequestration in *Pinus densiflora f. multicaulis*

The model itself was found to be significant at a probability of 0.000, and the factor model's explanatory power affecting the carbon sequestration of bounces was high at 95.8%.

The coefficient analysis revealed that only biomass had a significant impact on the carbon sequestration of the Pinus *Pinus densiflora f. multicaulis*, and that it was positively correlated with the amount of carbon sequestration. The regression equation is y = 0.013(Biomass) + 4.711 (Table 9).

**Table 9.** Analysis of factors affecting carbon sequestration in *Pinus densiflora f. multicaulis*.

| Model | Coefficients [a] | | | *t* | Sig. |
|---|---|---|---|---|---|
| | Unstandardized Coefficients | | Standardized Coefficients | | |
| | B | Std. Error | Beta | | |
| 1 (Constant) | 4.711 | 0.662 | | 7.113 | 0.000 |
| Biomass | 0.013 | 0.001 | 0.979 | 14.271 | 0.000 |

[a] Dependent Variable: Carbon sequestration. R Square = 0.958.

### 4.4.4. Tree Characteristics That Affect Carbon Sequestration in *Lagerstroemia indica*

The statistical analysis carried out to evaluate the factors that influence carbon sequestration in *Lagerstroemia indica* yielded significant results. The model itself was found to be statistically significant with a probability of significance of 0.042. Furthermore, the factor model, which explains the variables affecting carbon sequestration in the trees, had an explanatory power of 59.6%, indicating a substantial ability to account for variations in carbon sequestration.

Upon closer examination of the coefficient analysis, it was found that the only significant variable affecting the carbon sequestration of *Lagerstroemia indica* was the width of the tree crown. Interestingly, this variable showed a negative correlation with carbon sequestration. The regression equation that represents this relationship is y = −3.465(crown width) + 19.104 (Table 10).

**Table 10.** Analysis of factors affecting carbon sequestration in *Lagerstroemia indica*.

| Model | Coefficients [a] | | | *t* | Sig. |
|---|---|---|---|---|---|
| | Unstandardized Coefficients | | Standardized Coefficients | | |
| | B | Std. Error | Beta | | |
| 1 (Constant) | 19.104 | 4.958 | | 3.853 | 0.012 |
| Crown Width | −3.465 | 1.275 | −0.772 | −2.717 | 0.042 |

[a] Dependent Variable: Carbon sequestration. R Square = 0.596.

### 4.4.5. Tree Characteristics That Affect Carbon Sequestration in *Prunus yedoensis*

The statistical analysis conducted to identify the factors influencing the carbon sequestration of Yoshino cherry trees showed that the model itself had a significant probability of 0.036. Additionally, the factor model affecting carbon sequestration in Yoshino *Prunus yedoensis* was found to have substantial explanatory power, reaching a level of 54.7%.

Upon analyzing the coefficients, it was determined that only crown width had a discernible impact on the carbon sequestration of *Prunus yedoensis*. The study found a negative correlation between crown width and carbon sequestration, indicating that as the crown width increases, carbon sequestration decreases. The regression equation that represents this relationship is y = −2.932(crown width) + 26.538. These results highlight the significance of crown width in the carbon sequestration dynamics of *Prunus yedoensis* (Table 11).

**Table 11.** Analysis of factors affecting carbon sequestration in *Prunus yedoensis*.

| Model | Coefficients [a] | | | *t* | Sig. |
|---|---|---|---|---|---|
| | Unstandardized Coefficients | | Standardized Coefficients | | |
| | B | Std. Error | Beta | | |
| 1 (Constant) | 26.538 | 5.193 | | 5.110 | 0.002 |
| Crown Width | −2.932 | 1.0889 | −0.740 | −2.694 | 0.036 |

[a] Dependent Variable: Carbon sequestration. R Square = 0.547.

4.4.6. Tree Characteristics That Affect Carbon Sequestration in *Cornus officinalis*

The statistical analyses were conducted to identify the tree characteristics that influence carbon sequestration by *Cornus officinalis*. The model was found to have a high level of significance ($p$ = 0.000) and an impressive explanatory power of 87.2%.

The coefficient analysis revealed that biomass was the only significant variable affecting the carbon sequestration of the live trees. The study found that there is a positive correlation between biomass and carbon sequestration. This means that as biomass increases, carbon sequestration also increases. The regression equation that summarizes this relationship is y = 0.035(biomass) + 2.128 (Table 12).

**Table 12.** Analysis of factors affecting carbon sequestration in *Cornus officinalis*.

| Model | Coefficients [a] | | | $t$ | Sig. |
| | Unstandardized Coefficients | | Standardized Coefficients | | |
| | B | Std. Error | Beta | | |
|---|---|---|---|---|---|
| 1 (Constant) | 2.128 | 0.773 | | 2.751 | 0.025 |
| Crown Width | 0.035 | 0.005 | 0.934 | 1.369 | 0.000 |

[a] Dependent Variable: Carbon sequestration. R Square = 1.

4.4.7. Tree Characteristics That Affect Carbon Sequestration in *Chionanthus retusa*

In the analysis of *Chionanthus retusa*, the iterative linear approach did not yield significance, so a non-linear approach was adopted. The quadratic regression model, which had the highest probability of significance and explanatory power, was used for a more accurate representation.

The model of tree characteristics affecting *Chionanthus retusa* carbon sequestration is significant at the 99% level and has an explanatory power of 95.1%. The regression equation can be expressed as $y = 10.74 + 0.01 \times x - 1.13 \times 10^{-5} \times x^2$ (Table 13).

**Table 13.** Analysis of factors affecting carbon sequestration in *Chionanthus retusa*.

| Model | Coefficients [a] | | | $t$ | Sig. |
| | Unstandardized Coefficients | | Standardized Coefficients | | |
| | B | Std. Error | Beta | | |
|---|---|---|---|---|---|
| 1 Biomass | 0.014 | 0.003 | 1.229 | 5.156 | 0.001 |
| Biomass $\times x^2$ | $-1.135 \times 10^{-5}$ | 0.000 | -2.054 | -8.616 | 0.001 |
| (Constant) | 10.735 | 1.280 | | 8.387 | 0.001 |

[a] Dependent Variable: Carbon sequestration. R Square = 0.815.

It can be seen that the highest carbon sequestration for the *Chionanthus retusa* in the study area is between 500 and 1000. This suggests that while carbon storage capacity increases with tree growth, there is a threshold beyond which carbon sequestration may decrease, leading to a reduction in sequestration efficiency.

4.4.8. Tree Characteristics That Affect Carbon Sequestration in *Pinus densiflora*

The analysis of *Pinus densiflora* did not yield significant results using the repeated linear approach. Therefore, a non-linear approach was adopted, using a quadratic regression model with the highest probability of significance and explanatory power.

The model assessed the tree characteristics that influence pine carbon sequestration and showed significance at the 99% level, with an impressive explanatory power of 95.1%. The regression equation that represents this relationship is $y = 3.77 + 0.02 \times x - 2.38 \times 10^{-6} \times x^2$. This equation provides a nuanced understanding of the complex dynamics of carbon sequestration in pine trees (Table 14).

**Table 14.** Analysis of factors affecting carbon sequestration in *Pinus densiflora*.

| Model | Coefficients [a] | | | *t* | Sig. |
| --- | --- | --- | --- | --- | --- |
| | **Unstandardized Coefficients** | | **Standardized Coefficients** | | |
| | **B** | **Std. Error** | **Beta** | | |
| 1 Biomass | 0.017 | 0.001 | 4.440 | 16.191 | 0.001 |
| Biomass × $x^2$ | $-2.377 \times 10^{-6}$ | 0.000 | −4.165 | −15.190 | 0.001 |
| (Constant) | 3.771 | 1.290 | | 2.923 | 0.006 |

[a] Dependent Variable: Carbon sequestration. R Square = 0.883.

The study of *Pinus densiflora* in the area revealed a clear pattern: the highest carbon sequestration was observed in the biomass range of 3000 to 4000.

4.4.9. Synthesize Tree Attributes That Affect CS

The analysis revealed that crown width was the main factor affecting carbon sequestration in *Acer palmatum*, *Lagerstroemia indica*, and *Prunus yedoensis*, and an inverse relationship was found between them. In other words, the greater the crown width, the lower the carbon sequestration.

The rate of crown missing had an inverse relationship with carbon sequestration in *Quercus palustris*. Greater crown missing resulted in reduced carbon sequestration.

In *Pinus densiflora f. multicaulis* and *Cornus officinalis*, carbon sequestration increased with biomass.

For *Chionanthus retusa* and *Pinus densiflora*, the relationship between biomass and carbon sequestration was non-linear, as observed in more than 30 individuals studied. It is suggested that trees undergo a growth phase during which their biomass and carbon sequestration increase. However, after this phase, their capacity for carbon sequestration is expected to decrease. Therefore, it can be inferred that as trees age, their ability to take in carbon will decline.

## 5. Discussion

### 5.1. CSCS Depending on Tree and Regions

The study found that certain tree species, including *Salix pseudolasiogyne*, *Taxodium distichum*, *Ginkgo biloba*, *Quercus serrata*, *Aesculus turbinata*, *Pinus densiflora*, and *Pinus densiflora f. multicaulis*, have the highest carbon storage per tree. The study also found that larger trees have higher carbon storage and that biomass is closely related to tree diameter.

The study examined the carbon sequestration of individual tree species and found that *Salix pseudolasiogyne*, *Quercus serrata*, and *Quercus palustris* had the highest sequestration rates. Additionally, the study found that deciduous trees have higher sequestration rates than evergreens.

The study analyzed carbon storage and sequestration across 22 zones and found that the zones with the highest carbon storage were U15, U22, U8, U1, and U10. These zones contained large trees with significant biomass, crown diameter, width, and height, as well as a high number of trees. The sites with the highest sequestration were U22, U16, U1, U20, and U2, with U22 and U1 performing particularly well in both carbon storage and sequestration. A significant number of trees shared between both sites was a crucial factor contributing to these findings.

The study also found that carbon storage is higher in the G-E-E and G-E-D types, which are closed types with more than 70% vegetation cover in grassland areas. The G-E-E and G-P-E types, which have a higher proportion of evergreen trees than deciduous trees, have the lowest carbon storage to sequestration ratio.

### 5.2. Urban Green Space Planning Strategies for Enhancing CSCS

In this chapter, we provide guidelines for designing urban parks to maximize carbon storage and sequestration. We compare our results on differences in carbon storage and sequestration capacity by tree type with similar studies.

Pine trees are commonly utilized as ornamental trees in parks across Korea, including the area studied. However, findings indicate that pines are less efficient in carbon sequestration compared to other species, making them suboptimal for carbon sink purposes. This is supported by Han et al.'s research [50], which shows that pine plantations, due to their coniferous nature, are not effective carbon sinks because of their lower carbon storage and sequestration rates per area of vegetation. While pine and fir trees can be used for shade and aesthetic purposes in park designs, their overuse should be avoided.

The research also examined the carbon sequestration capabilities of *Chionanthus retusa* versus pine, noting a decline in sequestration as biomass increases. It was observed that biomass growth rates diminish 20 to 30 years post-plantation [51,52]. The study's outcomes indicate that areas with grasslands and dense vegetation cover above 70% exhibit greater carbon storage. Conversely, areas with a higher proportion of evergreen trees than deciduous ones show reduced carbon storage in relation to sequestration. Deciduous trees, due to their superior carbon sequestration capabilities, are recommended for sustained, long-term carbon sequestration, aligning with previous studies [53,54] that favor deciduous over evergreen species for carbon sequestration. The main reason is due to their natural growth patterns compared to other spaces [54].

Moreover, the presence of multilayered vegetation has been linked to enhanced carbon sequestration and improved ecological functions, including air pollution reduction, water cycle regulation, and biodiversity support [44,47]. These aspects should be integrated into urban park designs to maximize short-term carbon sequestration through an optimal mix of plantings. Beyond choosing tree varieties known for their significant carbon sequestration capacities and diverse planting schemes, it is crucial to focus on additional planting methods that can amplify the carbon sequestration capacity of urban green spaces. Kumar et al. [55] underscore that implementing practices such as conservation tillage, cover cropping, and reforestation efforts are instrumental in enhancing soil quality. These practices not only help in mitigating soil erosion and fostering biodiversity but also play a vital role in augmenting the soil and plant biomass's ability to store and absorb carbon.

It is important to highlight that existing studies suggest the configuration of landscapes significantly influences carbon sequestration capacity. According to [56], disruptions in the continuity of urban landscape patterns are associated with a decrease in carbon sequestration. Therefore, in urban planning, ensuring that newly created green spaces are integrated with and complement the existing green network not only enhances carbon sequestration but also contributes to greater biodiversity.

Enhancing carbon storage and sequestration efficacy involves not only increasing the variety and density of plantings in parks but also balancing these ecological goals with the aesthetic and functional needs of urban spaces. Designing parks that serve as enjoyable living spaces for city dwellers entails considering scenic value and ease of maintenance alongside environmental benefits [44,57].

To sum up, optimizing urban parks for enhanced carbon sequestration requires carefully choosing species of trees, leveraging the carbon sequestration potential of grasslands and dense vegetation, adopting layered planting strategies, and ensuring these initiatives complement the visual and practical needs of the parks. Utilizing insights from related studies and results from this study, urban planners and landscape architects can play a pivotal role in reducing urban carbon footprints, all while fostering lively and healthful community spaces.

### 5.3. Limitations of the Study and Future Research Prospects

This research has several limitations to discuss. First, the reliance on subjective evaluations of tree health and mortality rates during field surveys may introduce inaccuracies in

the measurements of carbon storage and sequestration. Second, the analysis was limited to selected sections of the park rather than its entirety. Third, when specific tree species did not align, calculations of carbon storage and sequestration using the i-Tree Eco model were based on the genus level. Future studies would benefit from broader sample sizes to enhance the precision of carbon storage calculations. Moreover, ongoing observation of tree growth in the various park types examined in this research is crucial to understanding the evolution of carbon storage and sequestration as trees mature. Adopting a long-term perspective will yield critical insights for the design of urban park landscapes with optimized carbon sequestration capabilities. Comparative analyses across multiple locations are also necessary for a comprehensive understanding. Furthermore, broadening the scope of carbon quantification to include vegetation beyond trees and incorporating soil studies may lead to more precise assessments of carbon storage and sequestration in urban green spaces.

## 6. Conclusions

This study provides new insights into the carbon storage and sequestration capabilities of urban green spaces. The study, which used a combination of advanced measurement techniques including the i-tree eco model, drone-based modeling, and on-site surveys, examined the park's different grid sections to understand the spatial dynamics of carbon storage and sequestration.

The study found that the average carbon storage across Yurim Park is 15.3 tons of carbon per hectare, with variations ranging from 5.0 to 21.4 tons per hectare depending on the area. The highest carbon sequestration and storage values were observed in areas dominated by broad-leaved trees and a closed canopy cover, highlighting the importance of plant area ratio and tree types on the park's carbon sequestration efficiency. The study's detailed area-based analysis offers practical insights into strategies and policies aimed at enhancing urban carbon sinks. By identifying the types of vegetation and specific configurations that maximize carbon sequestration, the study provides a guide for the development of sustainable urban parks that can serve as effective urban carbon sinks. Furthermore, the research methodology and findings provide a replicable model for similar studies in other urban settings, contributing to a broader understanding of the role of urban green spaces in mitigating climate change through carbon sequestration.

In conclusion, this study highlights the importance of strategic urban green space planning to improve carbon sequestration in urban areas. The findings emphasize the need for tree species selection, land cover management, and the optimization of green space configurations to enhance urban carbon sequestration capabilities. Urban green spaces play a pivotal role in achieving carbon neutrality, and this research underscores the necessity for integrated approaches that leverage the carbon sequestration potential of urban parks as part of broader climate change mitigation strategies.

**Author Contributions:** Conceptualization, J.K. and Y.K.; Methodology, Y.K. and E.J.K.; Investigation, J.K. and Y.K.; Software, J.K., S.S. and D.K.; Validation, Y.K. and E.J.K.; Formal analysis, J.K. and D.K.; Data curation, S.S.; Writing—original draft preparation, J.K.; Writing—review and editing, Y.K. and E.J.K.; Visualization, J.K., S.S. and D.K.; supervision, Y.K.; Funding acquisition, Y.K. All authors have read and agreed to the published version of the manuscript.

**Funding:** This research was supported by the basic research program through the National Research Foundation of Korea (NRF), funded by the Ministry of Education (Grant No. 2022R1I1A3056151).

**Data Availability Statement:** Data can be shared upon request.

**Conflicts of Interest:** The authors declare that they have no known competing financial interests or personal relationships that could have appeared to influence the work reported in this paper.

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
