# Peer review of "Carbon Storage and Sequestration Analysis by Urban Park Grid Using i-Tree Eco and Drone-Based Modeling"

_forests, doi:10.3390/f15040683_

Round 1
Reviewer 1 Report
Comments and Suggestions for Authors
The manuscript with the title “Carbon Storage and Sequestration Analysis by Urban Park Grid Using i-Tree Eco and Drone-Based Modeling”, conducted case study in Yulim Park, from Yuseong-gu, Daejeon, South Korea, with the purpose to asses carbon storage and sequestration. The purpose is to enhance ecosystem services provided by the vegetation from urban parks. Data was collected wit drones and subsequently underwent analysis and modelling.
I am not sure chapter 2 should be separate from the introduction.
Material and Methods are well-detailed.
Table 2 – name of species with italics; for some species the author is given (e.g Lines 1, 10, 12, 14) for some not. Please choose a format either give the author name or not. Also, please check the spelling on a reputable data base such as IPNI, to ensure full accuracy. If you choose to give the name of the author after the species, this should be done only when first time mentioned. Hence in table 3, these should not be repeated (e.g. Table 3 Lines 6 and 8).
Chapter 3.5. Scientific plant names with italic. Same for chapter 4.1. and Discussion section.
Best regards.
moderate English style improvements are needed.
Author Response
1. I am not sure chapter 2 should be separate from the introduction.
âž” Per your observation, the redundancy of content across the introduction and literature
review chapters negates the purpose of segregating them into distinct sections.
Nonetheless, recognizing the imperative to delve into the existing research concerning
factors influencing carbon sequestration, we opted to retain the current structure while
enriching the literature review framework. This approach underscores our commitment to
providing a comprehensive analysis by highlighting the significance of further exploration
with the theoretical domain, thereby justifying the maintenance of separate chapters for
introduction and literature reviews.
2. Table 2 – name of species with italics; for some species the author is given (e.g Lines 1, 10,
12, 14) for some not. Please choose a format either give the author name or not. Also, please
check the spelling on a reputable data base such as IPNI, to ensure full accuracy. If you choose
to give the name of the author after the species, this should be done only when first time mentioned.
Hence in table 3, these should not be repeated (e.g. Table 3 Lines 6 and 8).
âž” As you pointed out, we have corrected all parts related to species names.
3. Chapter 3.5 Scientific plant names with italic. Some for chapter 4.1 and Discussion section.
âž” Species names in the results and discussion have also modified
Please see the attachment.

Reviewer 2 Report
Comments and Suggestions for Authors
The study examines the carbon sequestration capacity of urban green spaces and give suggestions for planning effective urban parks in cities. They examine the carbon storage and carbon sequestration capacity of each tree species type using i-Tree Eco model and study the tree specific factors (crown size, canopy width, etc) affecting the sequestration capacity using regression models.
I have the following question and suggestions for the authors to further improve the manuscript:
1. The following content in lines 131-134 seems irrelevant or have been included by mistake: "Technical terms are explained when first used, and the language is clear, concise, and value-neutral. The text adheres to conventional academic structure and formatting, with consistent citation and footnote style. The grammar, spelling, and punctuation are correct, and the text is free from filler words and colloquial expressions.” – Please correct this.
2. The literature review in section 2.2 seems rather short. Can the authors please include other studies on what factors contributed to carbon sequestration for each tree type [Ex: Width, canopy cover, age etc.]. Currently it seems your regression analysis for few tree types discussed in section 4.4 is unique.
3. Also for section 4.4., please specify the number of trees used in regression calculation for each species.
4. In Table. 4, values in Carbon storage (kg) and Carbon sequestration (kg/yr) columns are same. Can the authors correct this?
5. The authors focus on the effectiveness of each tree for carbon sequestration, yet practical limitations for each tree types include water consumption, maintenance cost etc. Can these factors be considered for effective planning of urban parks to achieve sequestration goals with lower expense? I encourage the authors to comment or add to the discussion.
6. Can the authors illustrate on what are effective strategies to reduce field surveys for similar studies? Are maps of tree types available for the study area?
7. In lines 244-245, add citations for the methodology used to derived ground cover and crown height using DSM.
8. For Section 4.1, please explain the reasons for varying rates of carbon storage and sequestration for each species. Why trees with higher sequestration rates do not have high storage rates.
9. For the graph in Fig. 7, please annotate the y-axis appropriately for carbon storage (kg) and sequestration (kg/yr).
Author Response
- The following content in lines 131-134 seems irrelevant or have been included by mistake: "Technical terms are explained when first used, and the language is clear, concise, and value-neutral. The text adheres to conventional academic structure and formatting, with consistent citation and footnote style. The grammar, spelling, and punctuation are correct, and the text is free from filler words and colloquial expressions.” – Please correct this.
- Thank you for your thorough review of our manuscript. The issues you pointed out have been addressed and corrected.
- The literature review in section 2.2 seems rather short. Can the authors please include other studies on what factors contributed to carbon sequestration for each tree type [Ex: Width, canopy cover, age etc.].
- As you pointed out, part 2.2 was rewritten with further supplements. (page 3-4)
- Also for section 4.4., please specify the number of trees used in regression calculation for each species.
- Table 3 lists the methods used for analysis, tree species, and population numbers.
- In Table. 4, values in Carbon storage (kg) and Carbon sequestration (kg/yr) columns are same. Can the authors correct this?
- All figures have been modified to match the contents of the table.(page 9)
- 5. The authors focus on the effectiveness of each tree for carbon sequestration, yet practical limitations for each tree types include water consumption, maintenance cost etc. Can these factors be considered for effective planning of urban parks to achieve sequestration goals with lower expense? I encourage the authors to comment or add to the discussion.
- The empirical carbon sequestration strategy was rewritten by referring to existing studies and the results of this study. (page 19-20)
- Can the authors illustrate on what are effective strategies to reduce field surveys for similar studies? Are maps of tree types available for the study area?
- Since there was no separate tree map in the study area, satellite images and directly measured drone images were used.
- 7. In lines 244-245, add citations for the methodology used to derived ground cover and crown height using DSM.
- This part was not quoted from other literature, but was written directly by our research team.
- For Section 4.1, please explain the reasons for varying rates of carbon storage and sequestration for each species. Why trees with higher sequestration rates do not have high storage rates.
- This content has been supplemented by adding Figures 8 and 9. (page 11)
- 9. For the graph in Fig. 7, please annotate the y-axis appropriately for carbon storage (kg) and sequestration (kg/yr).
- All graphs have been modified.

Reviewer 3 Report
Comments and Suggestions for Authors
The research significantly contributes to understanding carbon dynamics in urban green spaces, vital for sustainable urban planning and climate change mitigation. To enhance clarity, distinguish between carbon storage and sequestration consistently. Improve coherence between tables and figures, ensuring accurate representation of data. Consider consolidating tables to improve readability in line with journal standards. Emphasize the practical implications of findings for policymakers and urban planners to maximize the paper's impact.
Comments and suggestions:
1. In Table 6, line 317, the last row, i.e., total carbon storage mentioned is confusing. Please check whether the total carbon storage and sequestration should be the same or not!
2. Based on Table 6, the data presented in Figure 6 are confusing. Please check it carefully for accuracy and consistency.
3. In Figures 7 & 8, the legend should be corrected. One of them should be labeled as "carbon sequestration."
4. Is it possible to merge the data from Tables 7 to 14 into a single table? Since this is a journal article, having too many tables may detract from the reader's attention. Consider consolidating the data to improve readability.
5. The author could customize the grid on the study area map by visually representing green spaces without color fill in the grid. Additionally, presenting the carbon storage and sequestration values for each grid directly on the map would enhance clarity and visualization.
6. Overall, the concepts of carbon storage and sequestration should be more clearly presented, as there remains some confusion in the way the data presented. The author should carefully review the entire data analysis and clearly present their findings to address any ambiguity.
Author Response
- In Table 6, line 317, the last row, i.e., total carbon storage mentioned is confusing. Please check whether the total carbon storage and sequestration should be the same or not!
- We've looked at all the numbers for carbon storage and sequestration and explained the differences between them. (page 11)
- Based on Table 6, the data presented in Figure 6 are confusing. Please check it carefully for accuracy and consistency.
- Table 6 and Figure 6 have been rewritten based on your comments.
- Based on Table 6, the data presented in Figure 6 are confusing. Please check it carefully for accuracy and consistency.
- The file contents were checked again, and Table 6 was modified to the average value by type. (page 11-12)
- 3. In Figures 7 & 8, the legend should be corrected. One of them should be labeled as "carbon sequestration."
- We modified based on your comment (page 11-12)
- Is it possible to merge the data from Tables 7 to 14 into a single table? Since this is a journal article, having too many tables may detract from the reader's attention. Consider consolidating the data to improve readability.
- We modified based on your comment (page 14-18)
- The author could customize the grid on the study area map by visually representing green spaces without color fill in the grid. Additionally, presenting the carbon storage and sequestration values for each grid directly on the map would enhance clarity and visualization.
- We attempted to improve the readability of the contents by adding Table 8 and 9.
- Overall, the concepts of carbon storage and sequestration should be more clearly presented, as there remains some confusion in the way the data presented. The author should carefully review the entire data analysis and clearly present their findings to address any ambiguity.
- This part was explained more clearly in the theoretical considerations section. (page 3-4)

Round 2
Reviewer 2 Report
Comments and Suggestions for Authors
My comments have been sufficiently addressed.
Author Response
Thank you for your thorough review of the manuscript.
Reviewer 3 Report
Comments and Suggestions for Authors
In the revised version of the manuscript, the author has made commendable efforts to address concerns related to data tables. However, regrettably, this has inadvertently led to the introduction of errors, thereby casting doubt on the accuracy and reliability of the subsequent graphical representations. Moreover, the redundancy of data observed in both tables and figures, exemplified by instances like Table 4-Figures 5 and 6, as well as Table 5-Figures 7, 8, and 9, poses a significant risk of confusing readers and impeding the clarity of the paper. Furthermore, the ambiguity surrounding Figures 10-12 adds further complexity to the situation. Additionally, the failure to consolidate tables for inferential analysis not only misses an opportunity to streamline the presentation but also jeopardizes the potential for misinterpretation. These critical issues necessitate immediate attention to uphold the manuscript's credibility and ensure its coherence and effectiveness.
Author Response
- In the revised version of the manuscript, the author has made commendable efforts to address concerns related to data tables. However, regrettably, this has inadvertently led to the introduction of errors, thereby casting doubt on the accuracy and reliability of the subsequent graphical representations.
- We agree with certain points you have highlighted. It was our anticipation that incorporating both tables and figures would enhance the manuscript’s readability for the readers. Nonetheless, should this integration result in confusion among readers, we believe modifications are necessary to address this issue.
- The redundancy of data observed in both tables and figures, exemplified by instances like Table 4-Figures 5 and 6, as well as Table 5-Figures 7, 8, and 9, poses a significant risk of confusing readers and impeding the clarity of the paper.
- For Table 4, and Figures 5 and 6, redundancy in content resulted in the deletion of Figures 5 and 6. Likewise, Figure 7 was removed. Yet, it is considered prudent to maintain the figures presently numbered 4 and 5 (originally 8 and 9), which were developed using GIS. The act of mapping carbon sequestration and storage, as outlined in lines 338-343, extends the capabilities of tabular data by aiding in the comparison of spatial arrangements and the interconnections between values.
- The ambiguity surrounding Figures 10-12 adds further complexity to the situation.
- Figure 10-12 has been deleted to reduce ambiguity.
- The failure to consolidate tables for inferential analysis not only misses an opportunity to streamline the presentation but also jeopardizes the potential for misinterpretation.
- We find it challenging to discern if this suggestion constitutes a critique of the entire manuscript or specifically addresses the regression analysis. Should the commentary pertain to the manuscript in its totality, it has been acknowledged. Conversely, in the context of regression analysis, merging analyses or tables that consolidate the carbon sequestration and carbon storage metrics for each tree into a single entity might lead to increased confusion for the readers.
